# Challenges with Assessing and Treating Pain in Research Primates: A Focused Survey and Literature Review

**DOI:** 10.3390/ani12172304

**Published:** 2022-09-05

**Authors:** Emilie A. Paterson, Patricia V. Turner

**Affiliations:** 1Department of Pathobiology, University of Guelph, Guelph, ON N1G 0C4, Canada; 2Global Animal Welfare and Training, Charles River, Wilmington, MA 01887, USA

**Keywords:** pain assessment, 3Rs, veterinary medicine, analgesia, macaque, animal welfare

## Abstract

**Simple Summary:**

It is crucial that research primates receive adequate pain treatment from ethical, animal welfare, and research-related perspectives. There is limited research on current pain management in research primates. A survey was administered to primate veterinarians (*n* = 93 respondents) to investigate a veterinary approach to pain recognition and alleviation as well as the pain management challenges that primate veterinarians may face. Survey results were used to guide a subsequent literature review on the topic. This review discusses current evidence and challenges in research primate pain management such as limited pharmacokinetic data and efficacy testing as well as a lack of validated pain assessment tools to recognize and evaluate pain in primates. Both the survey and literature review demonstrate gaps and challenges in primate pain management, and suggest science-based recommendations for improving current management guidance as well as future areas of research.

**Abstract:**

Research primates may undergo surgical procedures making effective pain management essential to ensure good animal welfare and unbiased scientific data. Adequate pain mitigation is dependent on whether veterinarians, technicians, researchers, and caregivers can recognize and assess pain, as well as the availability of efficacious therapeutics. A survey was conducted to evaluate primate veterinary approaches to pain assessment and alleviation, as well as expressed challenges for adequately managing primate pain. The survey (*n* = 93 respondents) collected information regarding institutional policies and procedures for pain recognition, methods used for pain relief, and perceived levels of confidence in primate pain assessment. Results indicated that 71% (*n* = 60) of respondents worked at institutions that were without formal experimental pain assessment policies. Pain assessment methods were consistent across respondents with the majority evaluating pain based on changes in general activity levels (100%, *n* = 86) and food consumption (97%, *n* = 84). Self-reported confidence in recognizing and managing pain ranged from slightly confident to highly confident, and there was a commonly expressed concern about the lack of objective pain assessment tools and science-based evidence regarding therapeutic recommendations of analgesics for research primates. These opinions correspond with significant gaps in the primate pain management literature, including limited specific pharmacokinetic data and efficacy testing for commonly used analgesics in research primate species as well as limited research on objective and specific measures of pain in research primates. These results demonstrate that there are inconsistencies in institutional policies and procedures surrounding pain management in research primates and a lack of objective pain assessment methods. Demonstrating the gaps and challenges in primate pain management can inform guideline development and suggest areas for future research.

## 1. Introduction

Research primates may undergo surgical and other invasive procedures for experimental purposes as well as periodically for veterinary care, which may result in pain [1,2]. To meet ethical obligations and ensure good animal welfare and quality of scientific data, prompt pain mitigation is necessary [3,4,5] and is required by research regulatory and oversight bodies [6,7,8]. Effective pain treatment is dependent on being able to recognize pain and assess its intensity [9,10,11]. While pain management has been an ongoing subject of research in human and veterinary medicine, pain management in research primates is less well studied and documented.

The International Association for the Study of Pain defines pain as “An unpleasant sensory and emotional experience associated with, or resembling that associated with, actual or potential tissue damage” [12]. Many factors can influence pain perception in animals, such as age [13], sex [14], and the environment [15], as well as past experiences with pain [16], and social ranking [17]. This demonstrates that pain is a subjective experience in animals as in humans, and emphasizes the importance of individualized pain assessment and treatment. Conversely, pain alters many physiologic, psychologic, and behavioural parameters that in turn can be used to identify if an animal is experiencing pain [6,9,18,19,20]. Some of these changes, such as weight loss, decreased food intake, altered mentation, and others, are not specific for pain and need to be interpreted in context [11], [21,22,23]. Among the most commonly used pain indicators in research primates are behavioural assessment, in which activity levels, locomotion, posture, and species-typical behaviours are assessed [6,20]. However, unless baseline measures and guidelines are provided, behavioural observations can be highly subjective [24]. It can also be challenging to directly assess pain in primates given that they are prey species and tend to hide signs of pain in front of human observers [25].

Pain management in human and veterinary medicine has continued to be an ongoing topic of research given that issues of inadequate pain mitigation are universal. Primate pain management practices are largely based on empirical evidence and given that primates are a nondomestic and nonverbal species, it can be difficult to recognize, assess, and treat pain effectively [9]. A survey conducted almost two decades ago in the UK examining the recognition and assessment of pain in laboratory animals demonstrated that there is a lack of objective pain assessment methods and that pain management practices are poorly reported [26]. A more recent survey on pain management in research animals also demonstrated that there are gaps in pain management due to the lack of evidence-based knowledge and that existing pain management protocols are generally not species-specific or discipline-specific [27]. These two surveys pertain to all laboratory species and there is limited knowledge of the current pain management practices for research primates and the views of primate veterinarians on this topic. In the process of developing a new Association of Primate Veterinarians (APV) primate pain mitigation position statement [28], there was interest in knowing more about routine practices for assessing pain. To address this issue, a cross-sectional anonymous online survey was developed on pain management in research primates and issued to members of APV and European Primate Veterinarians (EPV). The objective of the survey was to better understand: (1) pain assessment practices and institutional policies regarding pain assessment methods and treatments; and (2) primate veterinarian perceptions about pain management, including challenges and confidence levels in recognizing and treating primates in pain. From the survey, recurrent themes were used to conduct a literature search on primate pain treatment and management, specifically, looking for validated pain assessment tools for research primates as well as studies reporting analgesic pharmacokinetics and efficacy for common New and Old World research primates.

## 2. Survey about Primate Veterinarians’ Perspectives on Pain Management in Research Primates

### 2.1. Narrative Review Methods

The literature review was performed using the following databases: EBSCOhost (MEDLINE, Academic Search Premier, Ipswich, MA, USA), Elsevier ScienceDirect Journals, GALE ACADEMIC ONEFILE, Google Scholar, JSTOR, ProQuest (Science database), PubMed, SAGE Premier Journals, Scholars Portal Journals, SpringerLink, Web of Science (all databases), and Wiley InterScience Journals. Themes were identified from survey results and divided into 2 major topics: pain treatment and/or analgesics and pain assessment and/or recognition. Keywords were identified for both topics and entered into the databases in which all date ranges were included and sorted by relevance.

### 2.2. Survey Methods

APV is largely a U.S.-based not-for-profit organization of veterinarians who provide care for and oversee the health of multiple species of primates in research settings. EPV is an EU-based not-for-profit organization of veterinarians that also provides care for research primates. Both organizations deliver educational content to their members and promote the informal exchange of experiences, knowledge, and research data to facilitate ongoing professional development on primate medicine, care, and welfare. The survey was sent to members of both organizations.

A 21-item questionnaire was developed (see Appendix A). The APV Pain Assessment Subcommittee members reviewed the survey and the APV Board of Directors approved the final version. No identifying information was collected of participants and the survey was administered by APV association management personnel, with only aggregate data provided to the researchers. The survey was deemed exempt from REB approval because of this. Participants were informed before answering the survey that their participation was voluntary, all answers would be anonymous, no incentives were used, and they could choose to leave questions unanswered at their discretion. To be eligible to participate in this study, individuals had to be members of APV or EPV. The survey was conducted from 28 January 2019 to 15 February 2019. APV members received the survey by email through the APV listserve. EPV members received the survey through the EPV LinkedIn group.

### 2.3. Survey Data Analysis

Descriptive statistics were conducted using Microsoft Excel (version 2017) on all questions except open-ended questions. The final open-ended questions were only answered by 29% of respondents; non-response rates on questions ranged from 0 to 14, with an average of 87 (94%) responses per question. Percentages given for responses are from the total number of responses received. Participants working in two types of facilities were grouped as follows: if academia and a private research or government facility were reported, they were categorized as the latter. Participants with two positions (example: clinical veterinarian and facility director), were classified using the more senior position as the primary position. For the three questions relating to type of pain control drugs used for primates, responses that included name brands were classified according to the US Food and Drug Administration generic name. A Likert scale was used to assess confidence levels of respondents on a particular topic; a score of 1 indicating not very confident and a score of 5 indicating highly confident.

### 2.4. Survey Results

#### 2.4.1. Demographics of Respondents and Primate Species

A total of 93 members of APV and EPV responded to the survey, representing at least a 20% response rate, given that there is a membership overlap between these two societies. Eighty-four (90%) respondents were from the USA, two (2%) were from Canada, two (2%) were from Germany, and five (4%) were single responses from other countries (Barbados, the Netherlands, China, and Israel). Forty-one (44%) respondents worked in an academic setting and thirty (32%) in a private or contract research institution. Seventy-four (81%) respondents indicated that their main role was as a clinical, research, or attending veterinarian at their facility. Eighty-nine (96%) respondents worked with primates at the time of the survey and the majority with macaques (genus *Macaca*). More in-depth demographic data can be found in Table 1.

#### 2.4.2. Policies or Procedures for Pain Recognition in Research Primates

Respondents were asked if their animal ethics committee had a formal experimental pain assessment policy for research primates, and if so, whether the Standard Operating Procedures (SOP) were generic or specific to primates. Sixty (65%) respondents indicated that their institutional animal ethics committee did not have a formal pain assessment policy for primates (Figure 1). Twenty-five (27%) respondents indicated that their institution had a formal pain assessment policy for primates, fifteen (16%) had SOPs specific to primate pain assessment, and nine (10%) had generic SOPs related to pain assessment for all species housed in their facility. Respondents were also asked what type of pain assessment methods they used for primates. Eighty-five (91%) respondents reported using direct observation (e.g., cage side), thirty-two (34%) reported assessing pain indirectly (e.g., video camera), and four (4%) reported that the primates could not be closely observed at their facility due to their housing situation. Over 90% of respondents used general activity levels, food consumption, disuse or guarding of a body part, posture, and lameness to identify pain (Figure 2).

Respondents were asked who was responsible for assessing pain in primates at their facility and if responses to pain treatment were evaluated. Based on responses received, veterinarians and veterinary technicians were primarily responsible for conducting pain assessments at all facilities. Seventy-eight (84%) respondents indicated that primates are routinely monitored in the post-procedural period to evaluate the effectiveness of analgesia. Respondents were also asked how often unplanned top-ups of analgesics occur. Forty-five (48%) respondents reported that unplanned top-ups of analgesics occur sometimes and thirty-four (37%) reported that this rarely occurs (Table 2).

#### 2.4.3. Methods Used to Alleviate Pain in Research Primates

In terms of methods used to alleviate pain in primates, most respondents indicated that analgesic drugs were generally given. Eighty-six (92%) respondents reported using nonsteroidal anti-inflammatory drugs (NSAID), eighty-four (90%) used opioids, and eighty (90%) reported using topical/local anesthetics (Figure 3). Meloxicam was the most widely used NSAID and was reported to be used by 77 (83%) respondents (Figure 4A). Of these respondents, 13 indicated also using the sustained-release formulation of meloxicam. For opioids, 80 (86%) respondents reported using buprenorphine (Figure 4B), with 36 further indicating that they also used the sustained-release formulation of buprenorphine. In terms of topical/local anesthetic treatment, 70 (75%) respondents reported using bupivacaine and 69 (74%) respondents reported using lidocaine (Figure 4C). Other analgesic agents were also used, but less frequently than those indicated above (Figure 4).

The most common methods of non-pharmacologic care included acupuncture (4%, *n* = 4), hydrotherapy (12%, *n* = 11), and massage therapy (9%, *n* = 8) (see Figure 3).

#### 2.4.4. Quality of Pain Assessments in Research Primates

Respondents were asked to self-report their confidence level in recognizing and managing pain in primates vs. that of their associates. Forty-two (45%) respondents indicated that they were highly confident in recognizing and managing pain whereas forty-four (47%) respondents indicated that they were somewhat confident that research personnel at their facility could recognize pain in research primates (Figure 5).

Additional pertinent comments provided by participants included, “…use a multimodal approach, whenever possible”, “…I think we need a shift in pain management which focuses on pre-emptive analgesia and intraoperative analgesic methods”. “To me, postoperative analgesic protocols are well established but relied on too much”. and “Research staff are sometimes the first to pick up on subtle signs, but trained veterinary technicians are also very good at assessments”. “They are key components in the monitoring of all”. Finally, one respondent noted, “…unfortunately we still don’t have objective tools to score pain and the effectiveness of the provided analgesics”.

### 2.5. Discussion

This study summarized the results of 93 laboratory animal veterinarians, largely based in the U.S.A. across a range of employment sectors, and all with significant experience working with primates in research settings. The most significant finding of this survey indicates that primate pain management may be less than optimal due to inconsistencies in institutional policies and procedures and a lack of objective pain assessment tools in research primates. Macaques were the most reported species of primates housed from responding facilities, as expected from a 2014 survey on the use of research primates in North America in which rhesus and cynomolgus macaques were the most commonly reported species [29].

These survey results suggest that the majority of primate research facilities do not currently have a formal experimental pain assessment policy and even institutions that have a formal experimental pain assessment policy are not generally species-specific. A general survey on pain management in research animals demonstrated that there are inconsistencies in pain management across institutions and species as a consequence of not having specific guidelines in place [27]. Similarly, a survey conducted in the UK on pain recognition and treatment identified that only 6 of 25 institutions had a written policy for pain management [26]. The lack of formal pain management policies for research animals could be due to a lack of resources (i.e., time and scientific evidence) to effectively retrieve information and communicate pain assessment methods [24,26,30]. It may also be because there is a dearth of information published on the topic. As demonstrated in a number of recent literature surveys, researchers publishing in mainstream scientific journals have not been rigorous about reporting pain assessment and mitigation strategies, making it difficult for those searching for evidence of effective treatments [31,32,33,34].

The majority of respondents reported that veterinarians and veterinary technicians are largely responsible for conducting pain assessments at their facility and that these assessments are mostly conducted by direct observation. Rhesus macaques have been reported to suppress signs of illness following direct observation compared to indirect observation (i.e., video camera) [25]. Thus, exclusive use of direct observation methods may result in reduced detection of pain in primates. The predominant pain assessment technique reported was behavioural observation including general activity levels, disuse, or guarding of a body part, lameness, posture, and interactions with conspecifics [34]. As for health indicators of pain, food consumption and respiratory patterns were used most often as a measure of pain. All of the reported indicators are in line with the recommended guidelines for primate welfare assessments [28]. More recently, primate welfare assessment indicators have been identified using a Delphi method; although, the indicators are not specific to pain, rather they are indicators of general wellness and evaluate different categories of welfare including physical, environmental, and input-based measures [35,36]. As these guidelines state, a reference point (i.e., the individual’s normal behaviour) should be quantified so that when pain is assessed it is as objective as possible. Most indicators used are not specific to pain and need to be interpreted in context. Formalized score sheets are a good means to quantify pain behaviour, track frequency of evaluation, and can be kept in health records as a reference for the individual animal as well as the procedure.

Procedures that are thought to cause pain in humans need to be treated accordingly in animals unless proven otherwise. Pharmacologic methods were reported as the primary method of treatment. It can be difficult for veterinarians to choose which drug to use as well as the appropriate dosage due to the limited scientific evidence and possible research model pharmacologic restrictions [5,27,30,37]. It is a common practice in laboratory animal medicine to use multimodal pain treatment—that is, combining different drug categories to target different mechanisms of pain development [38,39]. In this survey, buprenorphine, meloxicam, and bupivacaine were the most commonly used opioids, NSAIDs, and topical anesthetics, respectively, for pain management in primates. These results are similar to the findings in a recent review of the analgesics and anesthetics reported in experimental surgical procedures in primates [31]. The only difference is in the NSAID category in which carprofen was reported more than meloxicam. Both meloxicam and carprofen have a similar mechanism of action and both are cyclooxygenase-2 selective [40]. Meloxicam has been demonstrated to be effective for postoperative pain mitigation in primates for orthopedic surgery [41,42] and neurosurgery [42,43] and is also used in combination with opioids [44]. A sustained-release formulation of meloxicam (0.6 mg/kg) is reported to result in therapeutic blood drug levels in cynomolgus macaques for 48–72 h compared to intramuscular or oral administration, which may last up to 24 h or 8–12 h, respectively [44]. Similarly, sustained release formulations of buprenorphine may last up to 96 h vs. 6–8 h for intramuscular formulations in macaques [45]. In the current survey, a small proportion of respondents reported using the sustained-release formulations of NSAIDs and opioids. It is unknown whether this is due to a lack of availability, concerns surrounding adverse side effects, a lack of knowledge about these formulations, or other reasons.

Perceived confidence can have an impact on pain management practices. This is a common phenomenon in human and veterinary medicine and recent surveys in both fields have queried levels of confidence in pain assessment and mitigation. Results from those surveys demonstrate that human nurses and veterinary technicians can have diminished levels of confidence due to limited knowledge on pain assessment and the appropriate analgesics to use, a lack of appropriate tools to assess pain objectively, and a lack of continuing education [27,30,37,46,47,48]. These factors also may have an impact on the confidence levels reported in this study, in which approximately half the respondents reported being somewhat confident in assessing and managing pain and the other half reported being highly confident. In a recent survey similar to this study, primate veterinarians were asked about their level of confidence and to associate their level of confidence with certain statements [49]. It was found that primate veterinarians who have higher levels of confidence will be more likely to use behaviour and facial expressions as pain indicators and to opt for an increased use of pain medication [49]. Conversely, in the current survey, when asked to report the perceived level of confidence in research personnel, the majority reported that they were only somewhat or less confident. We need to consider that confidence levels do not reflect skill level and that this survey assessed the participants’ self-reported confidence and not their objectively assessed ability to identify and evaluate pain in primates.

Study limitations include a small sample size, and thus, the results may not be reflective of the views and opinions of all APV and EPV members. Furthermore, the majority of respondents were from the U.S.A, and may not be reflective of the views and opinions of primate veterinarians in other countries. Finally, due to participant anonymity, answers could not be linked to the participant demographics, and thus it was not possible to assess the relationship between these parameters (for example, years of clinical experience vs. confidence in recognizing and treating pain in macaques).

Subsequent to the administration of this survey, the APV Guidelines on Pain Management were published [50]. It would be interesting to conduct a follow-up survey study to examine the impact of APV guidelines on pain management practices in different institutions. As demonstrated by this survey, there is a lack of objective pain assessment tools in research primates, and thus future research should focus on validating pain assessment tools for these animals.

## 3. Pain Assessments in Research Primates—A Review

To provide effective pain treatment it is important to recognize and evaluate the intensity of pain. There are various assays used for pain identification and assessment that fluctuate in objectivity, reliability, and practicality. Pain can be difficult to identify and quantify in primates as they are prey species and often hide signs of pain in front of human observers, unless severe [25]. In this section, we will present commonly used pain assessment methods or assays employed to evaluate pain in primates and discuss their pros and cons in research settings (see Table 3).

### 3.1. Reflex-Based Assays

Reflex-based assays are commonly used to evaluate dose-related increases in pain threshold to assess the efficacy of analgesics, but they are rarely used in clinical practice [64]. The approach usually involves the application of a standard noxious stimulus (i.e., chemical, thermal, or mechanic) followed by quantification of the animal’s reflex response. For example, a study using squirrel monkeys performing an operant behaviour (i.e., pulling on a thermal rod that increased in temperature) demonstrated that administration of several opioids resulted in dose-related increases in temperature threshold [52]. Similarly, in rhesus macaques, thermode behavioural testing has been validated pharmacologically as a tool to determine analgesic efficacy with commonly used opioids (i.e., tramadol, morphine, and fentanyl) [51]. This assay is a simple method to conduct and can be attributed a value. However, this type of assay only captures the sensory component, more specifically, hypersensitivity of the nociceptors; thus, this method does not capture the learned and emotional components of pain [53]. This suggests that the clinical significance may be poor, but these assays may be useful for the preliminary determination of the therapeutic efficacy of novel analgesics.

### 3.2. Physiologic Parameters

Pain causes a cascade of physiologic events that can be measured and quantified. Acute pain and breakthrough pain activate the sympathetic nervous system resulting in increased blood pressure, respiratory rate, body temperature, and heart rate [54]. Pain also activates the hypothalamic–pituitary–adrenal system causing an increase in certain hormones in serum (i.e., cortisol and adrenocorticotropin) [65]. Physiologic parameters are objective; however, there are currently no predefined, validated values that are specific to pain in primates. Moreover, measuring these parameters usually requires the capture and restraint of primates, which can skew the results by increasing arousal and stress [61]. Field research has made use of techniques to attempt to measure these parameters remotely, such as imaging photoplethysmography to measure heart rate from a distance [55]. Other possibilities include measuring cortisol levels in saliva, urine, or feces [18,56,57,58]. For example, Salimetrics^®^ Oral Swabs were validated for cortisol measurement in marmosets, in which animals are given a swab to chew (saliva can be extracted from the swab) [66]. Physiologic measures are not specific to pain and need to be interpreted using other measures and information.

### 3.3. Clinical Indicators

Clinical signs are representative of the outcome of animal care. In a research setting, body weight or body condition scores are recorded for study purposes but also to evaluate overall animal health. When body weight or body condition scores drop this usually signals a need for a further veterinary exam. Weight loss or a decrease in body condition score is an indirect measure of pain that may reflect a behavioural change related to pain; however, it can also be linked to other sources such as illness (i.e., chronic disease, cancer, or infection), social conditions, and the environment [67]. Body condition scoring may be a preferred measure in primates since it can be performed cage side. Currently, validated body condition score scales have been validated for macaques and chimpanzees [59]. However, there is no literature on body weight or body condition score changes in relation to acute or chronic pain in research primates. This method should be used as an indirect indicator of potential chronic pain, general health, or as a human endpoint [60].

Similar to body weight, food intake or appetite are clinical indicators related to general health. In a research setting, food consumption or food evaluations are generally values recorded for study purposes and for health monitoring [68]. When an animal experiences acute or chronic pain, it may lead to a reduced appetite [69]. It is important to note that both body weight and appetite will most likely be reduced in the days following a moderate to highly invasive procedure; thus, these values need to be interpreted in context [9]. These types of clinical measures are representative of a long duration of negative states, such as pain, and they should be used in conjunction with other measures of pain, to avoid prolonged welfare compromise. Usually when pain is anticipated or present animals are treated with analgesic agents. The administration of analgesics or anesthetics alone can also result in diminished food intake and body weight demonstrating the importance of context and variables that can influence these two clinical signs [70].

### 3.4. Behaviour

Cage-side behavioural observations are the most commonly reported wellness assessment in research primates [71]. Many welfare frameworks and uni/multidimensional scales have been created to quantify pain and behaviour. For example, the Extended Welfare Assessment Grid (EWAG) for the assessment of welfare and cumulative suffering in experimental animals evaluates the following components: clinical condition, experimental/clinal events, environment, and behavioural deviations [72]. Having a grid with specific welfare descriptors aids in assessing objective behavioural measures. However, this tool is not specific to pain assessment and there are currently no pain assessment tools for primates. Similarly, primate welfare assessment indicators have been identified and developed into a tool to assess general primate welfare in a research setting [35,36]. Again, these indicators are not specific to pain, but can be influenced by pain or pain treatment, for example, reduced appetite [35]. Another example more specific to pain is the Melbourne Pain Scale used in a clinical setting for cats and dogs. This tool assesses mostly behaviour indicators of pain, including activity levels, vocalizations, response to palpation, posture, mental status, and physiology measures [19]. These tools demonstrate that it is possible to measure behaviour objectively; however, as pain is a subjective and transient experience, knowledge of the animal’s normal state is necessary. This can be challenging in a research setting if regular behaviour and temperament assessments are not conducted or recorded. This demonstrates the importance of communicating with technical staff given their close, daily experience with animals.

Various behaviours can be indicative of potential pain in primates. General activity levels can be assessed to detect potential pain. A recent study examining wellness indicators in rhesus macaques in the post-procedure period found a significant reduction in overall activity levels and a decrease in the behavioural repertoire including arboreal behaviours (i.e., climbing, hanging, and standing up straight) [20]. Another behaviour indicative of potential pain found in the latter study was slouched posture, in which the head is positioned below the level of the shoulders. These findings are similar to a study that evaluated the efficacy of different analgesics following abdominal surgery in olive baboons, supported by telemetry data [43]. As with other pain indicators, decreases in general activity and hunched posture may occur due to other states, such as depression [63]. Furthermore, behaviours may be influenced by the location and type of pain. Assessing guarding and disuse of a body part, vocalization in response to touch or palpation, as well as lameness, may be indicative of pain in a specific area. This demonstrates that pain should be interpreted in context and emphasizes the importance of keeping an observation log even when primates are not on study.

Changes in species-typical behaviour relative to baseline can also be indicative of potential pain. Primates are social species, thus evaluating social behaviour with conspecifics or humans can be helpful. In a worksheet developed to assess behaviour as a quality of life assessment in nonhuman primates, researchers incorporate several social behaviours such as affiliate behaviour (i.e., grooming, huddling, embracing, or proximity to others), play behaviour (i.e., wrestling, pulling, tickling, chasing, or play biting), aggressive behaviour (i.e., threatening, chasing, hitting, attacking, fighting, or biting), submissiveness to other (i.e., pant-grunting, lip-smacking, bobbing, avoiding, crying, or grimacing), and interest in a novel situation that includes humans [73]. Using a primate’s natural daily activity budget as a benchmark for measuring welfare can be useful. For example, primates spend ~40 to 60% of their daily activity budget foraging for food; thus, a significant alteration in this time proportion can be indicative that the animal is feeling unwell [74]. This study also emphasizes the importance of creating an environment that provides primates with the opportunity to perform species-motivated behaviours so that changes can be used as a measure of welfare and, potentially, be indicative of the presence of pain.

To quantify daily activity budgets, detailed ethograms can be created [75,76]. In the context of pain, ethograms can be helpful to quantify the reduction in normal or species-typical behaviour and the appearance of pain-related behaviours [77]. Software systems such as Observer XT can facilitate behavioural scoring and statistical analysis; however, conducting these assessments is laborious, inter- and intra-observer reliability needs to be assessed, and assessments need to be conducted in real time to be useful. To attempt to address the problem of real-time scalability, new technology is being developed to automate pain behaviour recognition for some species, such as mice [77,78,79]; however, none currently exist for research primates.

### 3.5. Facial Expressions

In the past decade, facial grimace scales have been developed for many species, including domestic [80,81], agricultural [82,83,84,85], and research [86,87,88] animals. The facial action units within these scales are similar among species, generally focusing on the eye, mouth, and nose areas. Generally, facial grimace scales are composed of 3–5 facial action units scored on a numerical scale resulting in a score that reflects the level of pain an animal is experiencing, ideally resulting in prompt decisions for pain treatment.

Grimace scales are relatively simple to conduct and a rapid measure of pain that is readily available and requires minimal training; however, this method should be used alongside other pain assessments to offer a holistic approach and make the most accurate decision for pain treatments. Although no validated grimace scale exists for primates, as reported in the survey (see Section 2), this method is used by primate veterinarians. In a recent study conducted on macaques, it was found that eye narrowing and lip tightening were present in the postoperative period compared to baseline [20]. However, due to the methodology and levels of variations between subjects these changes were not significant [20]. This shows that there may be similarities with other species and that future research may yet validate a facial grimace scale for research primates.

Facial grimacing is a novel measure of pain in animals and past research has focused on scoring images retrospectively. Currently, research is emphasizing the use and validation of the grimace scale for real-time use. The mice and rat grimace scales have been validated for cage-side use, and this has helped to demonstrate that recommended analgesic doses for certain procedures in mice and rats were insufficient, leading to new recommendations [61,89,90,91].

As with the other pain assessment methods, there are some considerations when using facial expressions to evaluate pain. Human observers can influence an animal’s facial expressions; thus, when possible, indirect observations are preferred in primates [25]. In a recent review, the main barriers to the widespread clinical application of grimace scales are discussed [92]. This includes not being able to compare the suspected painful animal to their baseline (i.e., the methods used to create grimace scales do not lead to practical implementation); statistical significance in parameter changes may not translate to clinical significance, and thus it is important to set an intervention threshold; and the variance between observer and their experience with a given species, emphasizing the importance of having a robust training [92]. Grimace scales have great potential for clinical use, but need to be used alongside clinical assessment and behaviour.

## 4. Pain Treatments in Research Primates—A Review

The standard means to treat pain in research primates is through pharmacologic methods. Most therapeutic pain treatments fall into one of the following classes: opioids, nonsteroidal anti-inflammatory drugs, and local anesthetics. In this section, we present the challenges of treating pain in research primates, the primary analgesics used with pharmacokinetic values and efficacy evidence (where available), and the routes of administration and potential side effects.

### 4.1. Research Primates and Analgesics

Treating pain in research animals can be challenging due to the balance between scientific outcome (i.e., analgesic interaction with test articles or models and the confounding effect of unalleviated pain) and animal welfare (i.e., ethical duty to minimize pain and distress) [5]. There is a lack of evidence-based knowledge for pain management in primates specifically for pharmacokinetics and analgesic efficacy as well as a lack of reporting detailed analgesic protocols [31,32].

Consequently, current recommendations are often extrapolated from other species, even though pharmacokinetics can have significant interspecies and intraspecies variability. To effectively treat pain, it is necessary to know the appropriate analgesic, dosage, frequency, and period of action. Furthermore, pain is a subjective experience and varies in intensity and duration depending on the individual’s sex [93], previous experience with pain [16], psychological state [94], social status [17], and genetics [95]. Considering this variability, pain treatments should be tailored to the individual, which can be difficult in a research setting when working with large groups of animals.

### 4.2. Opioids

Opioids are generally used to treat moderate to severe anticipated pain in research primates. As reported in Section 2, the most commonly used opioids are buprenorphine, hydromorphone, fentanyl, and tramadol. These opioids have various mechanisms of action and potencies. For example, buprenorphine is a partial mµ agonist and can reach a plateau, limiting analgesic effects, whereas fentanyl is an mµ agonist and serum concentrations rise as dosages increase [96]. These properties should be considered when developing a pain management protocol.

The most studied analgesic therapeutic class in research primates, in terms of pharmacokinetics, is opioids (see Table 4). When comparing the efficacy and pharmacokinetics reported in other laboratory species, such as rodents, empirical evidence in primates is very limited [97]. Another gap in the primate literature is efficacy testing. Most rodent studies conduct efficacy testing with reflex or chemical-based assays [97], whereas there are few validated reflex-based assays for research primates.

Studies evaluating efficacy testing in primates have usually assessed behaviour and physical indicators of pain in the post-operative period, but rarely report pharmacokinetic values. For example, the efficacy of buprenorphine (0.01 mg/kg given every 12 h) was assessed in olive baboons undergoing a surgical procedure with and without combination with carprofen [43]. Some individuals had elevated heart rates and reduced activity levels in the post-operative period when compared to the multimodal approach potentially indicative of pain [43]. Although this study did not report pharmacokinetic values, it can be interpreted with other studies that used the same dosage of buprenorphine in primates. For example, a study in cynomolgus macaques demonstrated that buprenorphine (0.01 mg/kg) had a half-life of 2.6 ± 0.7 h and a Cmax of 8.1 ng/mL with a recommended dose interval of 6–8 h [45]. This study also examined the pharmacokinetics of buprenorphine at 0.03 mg/kg with a very high standard deviation (i.e., Cmax 40.7 ± 48.7 ng/mL) indicating high individual variability with the same dosage [45]. The half-life at this dosage was 5.3 h—suggesting that waiting 8–12 h before redosing may leave the animal without sufficient analgesic coverage. This emphasizes that more research that examines both efficacy and pharmacokinetics is needed in research primates.

Similarly, there is evidence of high variability for efficacy and pharmacokinetics between species. A recent study examining two formulations of transdermal fentanyl patches in cynomolgus macaques based on reported doses in dogs demonstrated adverse effects at 2.6 mg/kg, which had been demonstrated to be effective and safe in dogs [98]. This highlights some of the challenges in creating adequate pain management protocols in primates.

### 4.3. Nonsteroidal Anti-Inflammatory Drugs (NSAID)

NSAIDs are frequently used to treat mild to moderate pain in primates or are added as part of a multimodal regimen. Typically, NSAIDs act on cyclooxygenase-1 (COX-1) and COX-2 receptors to control the inflammatory response and provide analgesia [95,103]. As reported in section A, the most used NSAIDs in research primates are meloxicam, carprofen, ketoprofen, and flunixin. Through this review, it was determined that of these commonly used NSAIDs, only meloxicam has pharmacokinetic values specific to research primates and only carprofen has been studied for efficacy in primates (see Table 4). No pharmacokinetic studies are available for the other common analgesics reported in Section 2 (carprofen, ketoprofen, and flunixin); thus, data from other species are provided in Table 5. As mentioned, most information about dosages and dosing intervals for research primates contain values extrapolated from other species, such as rodents, cats, and dogs [60]. Oral and injectable preparations of both meloxicam and carprofen are available and recommended to be given once a day [104,105]. However, research in rodents shows that previously recommended doses of NSAIDs, such as carprofen were insufficient for common surgical procedures, demonstrated by elevated facial grimace scores [106,107]. This highlights the importance of pain recognition and assessment outside of reflex-based assays and the need to evaluate pain when using recommended doses of analgesic agents. Furthermore, as mentioned above, there is interspecies variability; thus, analgesics need to be used with care in primates with effects monitored frequently.

### 4.4. Multimodal Analgesia

Combining different classes of analgesic agents to target different pain pathways is often beneficial [39,114]. Although there are limited data on the pharmacokinetics of multimodal regimens in primates, it is known that when an opioid, an NSAID, and a regional block with a local anesthetic are combined, this allows for the reduction in the dosages/frequency of the individual drugs and consequently their side effects (multimodal analgesia regimes in rodents, reviewed in [96]). This review did not evaluate local anesthetics in depth; their use was queried in the survey in Section 2, and the most used local anesthetics were reported to be bupivacaine, lidocaine, proparacaine, and EMLA^®^ (a prilocaine/lidocaine mixture) cream. Local anesthetics, such as bupivacaine and lidocaine typically have a short duration of action of 60 min and 30 min, respectively, and are used around the surgical site to reduce peripherical nociceptor activity [104,115]. Thoughtful perioperative planning, for example, by administering NSAIDs prior to surgery, as well as opioids and local anesthetics during surgery to treat pain before its onset, has demonstrated a faster recovery, minimizing the potential for breakthrough pain [116]. More information is needed to optimize perioperative analgesia protocols in research primates.

### 4.5. Route of Analgesic Administration

There are many factors to consider when choosing the route of administration of therapeutics in primates, such as the stress associated with handling or immobilization, the frequency of administration to achieve therapeutic levels, the level of absorption/bioavailability, and the desired effect. Below we discuss the common routes used in primates as well as the advantages and disadvantages from a practical and physiological standpoint.

Oral administration of analgesics is common and some primates will voluntarily take the medication cage side if it is palatable. This is especially true when positive reinforcement training is employed with primates [117]. Another option for voluntary consumption is through chewable commercially available tablets. Common opioids used in research primates such as tramadol and hydromorphone as well as NSAIDs such as meloxicam, carprofen, ketoprofen, and flunixin are available in commercially produced oral preparations [96]. Oral administration may cause more efficacy variability compared to other routes of administration based on fed or fasted conditions of animals [118]. Furthermore, some therapeutic agents may irritate the gastrointestinal mucosa when given orally [59].

Subcutaneous (SC) injections are given between the layer of skin and muscle and can be administered over a large portion of the body. Typically, the rate of absorption is slower, which may be desirable for prolonged action [105]. Subcutaneous injections can cause depot accumulation; thus, injection sites need to be changed when multiple injections are given [105]. In primates, common opioids such as buprenorphine, hydromorphone, and fentanyl and NSAIDs such as meloxicam have been reported to be used SC [45,51,99,101,119].

Intramuscular (IM) injection is the most common route of analgesic administration in primates because it is an easy technique that requires minimal handling and restraint. IM injections are given in deep muscle tissue and the high vascularisation permits rapid absorption [105]. The volume per injection site should be limited based on the primate’s weight to minimize the potential for injury and necrosis. For example, a primate weighing approximately 3 kg or 13 kg should receive no more than 0.5 mL or 1.0 mL per site, respectively [120]. The standard IM injection sites for primates include the caudal thigh, the deltoids, and the longissimus (paralumbar) muscles to avoid major blood vessels and nerves [120]. Opioids such as buprenorphine and hydromorphone as well as NSAIDS, including carprofen and meloxicam, have been reported to be used via IM injection in research primates [43,44,45,121].

Intravenous (IV) injections are usually given in a superficial vein with a needle or via continuous infusion with a catheter. The rate of infusion/administration is controlled for a given time and smaller doses are generally required since agents are administered directly into the bloodstream [105]. Without sedation and/or training, primates may be unwilling to cooperate for a long duration. In primates, common opioids such as buprenorphine, hydromorphone, and tramadol have been administered through IV injection [51,72,101].

Finally, transdermal drug delivery via patch or other protected depot can be beneficial for long-term pain treatments through slow epidermal absorption and requires minimal handling after the first application [122]. Primates usually require a jacket (which can require further pre-study habituation) to avoid self or partner ingestion. Inadvertent ingestion of a fentanyl patch with fatal consequences has been reported in primates [123]. More recently, research on the use of transdermal fentanyl solution and patches has demonstrated prolonged serum concentrations at therapeutic levels over 3–5 days [97,100,102].

### 4.6. Adverse Effects

The goal of therapeutic pain treatment in animals is to create a balanced state and minimize the pain experienced without producing substantial adverse effects. Multimodal approaches are recommended when possible, as these techniques generally result in reduced dosages and dosing frequency when compared to individual analgesics. Each class of therapeutic analgesics has side effects based on their different action mechanisms, chemical structures, formulation pH, etc. It is important to know these side effects when treating pain to recognize them and modify treatment accordingly to ensure that a toxic level is not reached. From a research perspective, knowledge of adverse effects can be used to distinguish analgesic effects from test article effects or study outcomes. There is limited research into analgesic adverse effects in research primates; thus, these are mostly identified based on the literature for other species, such as dogs.

Opioids are potent drugs with a narrow window for therapeutic safety; consequently, they must be used with caution and doses must be calculated for the individual animal [104]. Opioids have been reported to cause respiratory depression, bradycardia, and when administered at high doses or via IV injection can cause hypertension [124]. Opioids can also affect the gastrointestinal tract by reducing mobility and emptying as well as inducing nausea and vomiting [125,126]. Opioids may induce behaviour changes, specifically, sedation. For example, in a study comparing the behavioural and physiologic effects of morphine versus fentanyl in dogs, significantly higher sedation scores were seen when fentanyl was the chosen analgesic [127]. The primary adverse effects of NSAIDs occur in the gastrointestinal tract, inducing ulceration, perforations, diarrhea, vomiting, and reduced appetite [128]. In extreme cases in dogs, some COX-2-specific NSAIDs have also been reported to cause hepatic failure and lethargy [129]. These side effects must be assessed at the individual level as different animals will react differently.

Different routes of administration can also create adverse effects. Skin puncture can be mildly painful, and it is important to regularly check injection sites as some animals may have an adverse reaction. For example, a small proportion of cynomolgus macaques injected SC with meloxicam SR showed adverse injection site reactions including redness, sloughing of superficial tissue, and abscess formation, whereas other animals in the same study did not [44]. If injections are frequent, consider recording injection sites to ensure that specific sites are not over-used to avoid tissue damage. If injections cause a severe reaction consider an alternate route of administration.

Distinguishing pain behaviour from sedation behaviour in research primates is important as signs can be similar. Furthermore, in some circumstances primates are immobilized with anesthetics such as ketamine to be manipulated; thus, when administered in conjunction with analgesics, side effects may be difficult to distinguish. The main side effect that anesthetics and analgesics have in common is reduced appetite. A study in rhesus macaques and African green monkeys evaluated the association between ketamine injections (10 mg/kg) and appetite, 24, 48, 72, and 120 h post-injection. The researchers demonstrated a significant decrease in food intake at all timepoints with 24 h post-dose being the most significant (mean % intake reduction: African green monkeys: 57%; rhesus males: 48%; rhesus females: 40%, respectively) [70]. Decreased food intake has also been reported following the use of analgesics in healthy animals, likely due to sedation [39].

## 5. Recommendations and Considerations for Refinement of Pain Management Guidance for Research Primates

### 5.1. Institutional Policy to Implement Pain Management Guidance

Based on our survey results (Section 2), there is evidence of the need to implement guidance within and between research institutions on primate pain assessment, pain treatment, and general pain management procedures. The Canadian Council on Animal Care (CCAC) has released new guidelines encouraging research facilities to create and implement welfare assessments for laboratory species including the need to incorporate indicators related to pain assessment and management [130]. There is a critical need for pharmacokinetic and efficacy testing (based on objective pain assessment methods) to inform treatment protocols for research primates as most evidence stems from anecdotal evidence.

### 5.2. Analgesic Administration Based on Empircal Evidence

There are many possible routes of administration for therapeutics, and it is important to consider the required handling/restraint for each method. Incorporating slow-release formulations may reduce the need for handling and restraint. These formulations also help to minimize the risk of breakthrough pain [131]. Given the variability of absorption between different dose routes, there should be frequent monitoring in the immediate hours following presumed pain with the administration of analgesics to ensure adequate pain relief.

### 5.3. Appropriate Use of Pain Assessment Tools

Research institutions should have a standardized pain assessment protocol that integrates two or more methods identified in Table 5. These protocols should have an objective scoring system that can be replicated by multiple users and that demonstrate consistent results over time. Integrating an analgesic threshold associated with the outcome of pain assessment should be established. The gold standard pain assessment method for research primates is indirect observation of behaviour. Combining this with other methods, such as physiologic and clinical markers, ensures a more reliable assessment of pain and thus better management and mitigation.

## 6. Conclusions

Results from a survey administered to primate veterinarians demonstrated inconsistencies in research primate pain management as well as a general lack of objective pain assessment tools. Information in this review may be used by research institutions to evaluate primate care as well as for creating primate-specific internal guidance. These inconsistencies correspond with gaps in the research primate pain literature, which includes limited pharmacokinetic studies and efficacy testing for commonly used analgesics as well as limited objective measures of pain. These findings should encourage researchers and veterinarians to study and report more detailed methods of pain management practices to further improve research primate welfare and the quality of scientific data.

## Figures and Tables

**Figure 1 animals-12-02304-f001:**
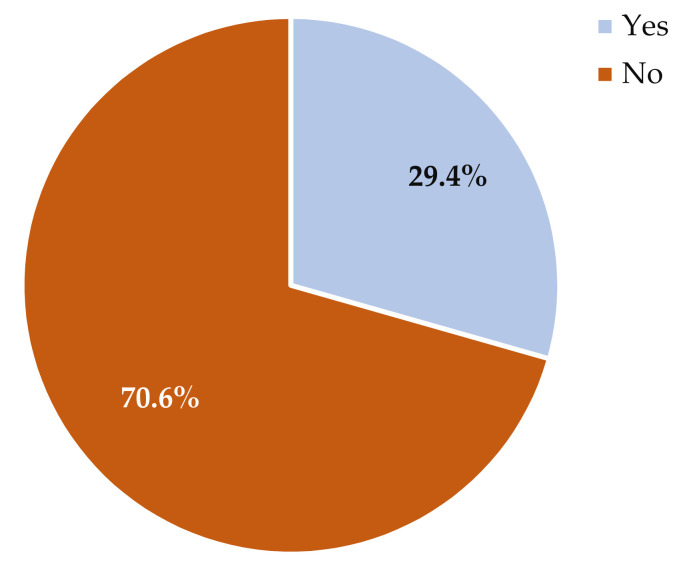
Percentage of animal ethics committees that have (blue) or do not have (red) a formal experimental pain assessment policy for research primates at their institution (*n* = 85).

**Figure 2 animals-12-02304-f002:**
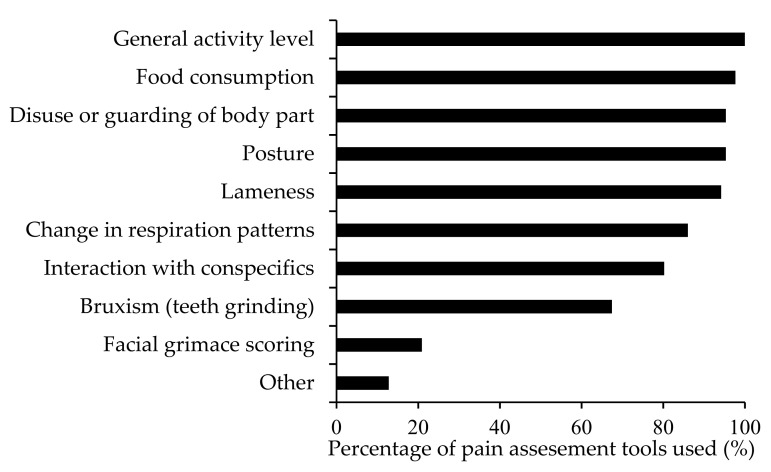
The different pain assessment tools in use (expressed as a percentage) for primates at the respondents’ facilities (*n* = 86).

**Figure 3 animals-12-02304-f003:**
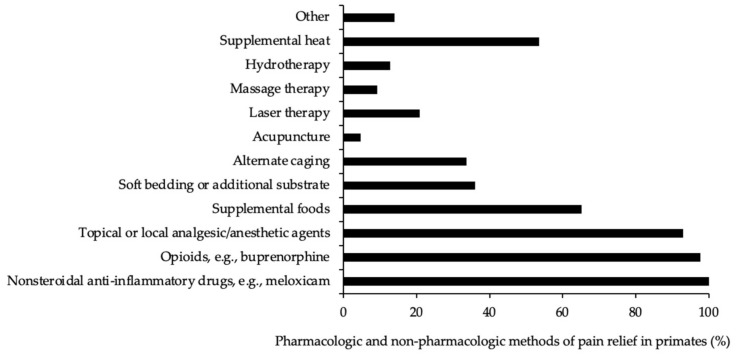
Methods used to treat and manage pain in research primates (expressed as a percentage of responses) (*n* = 86).

**Figure 4 animals-12-02304-f004:**
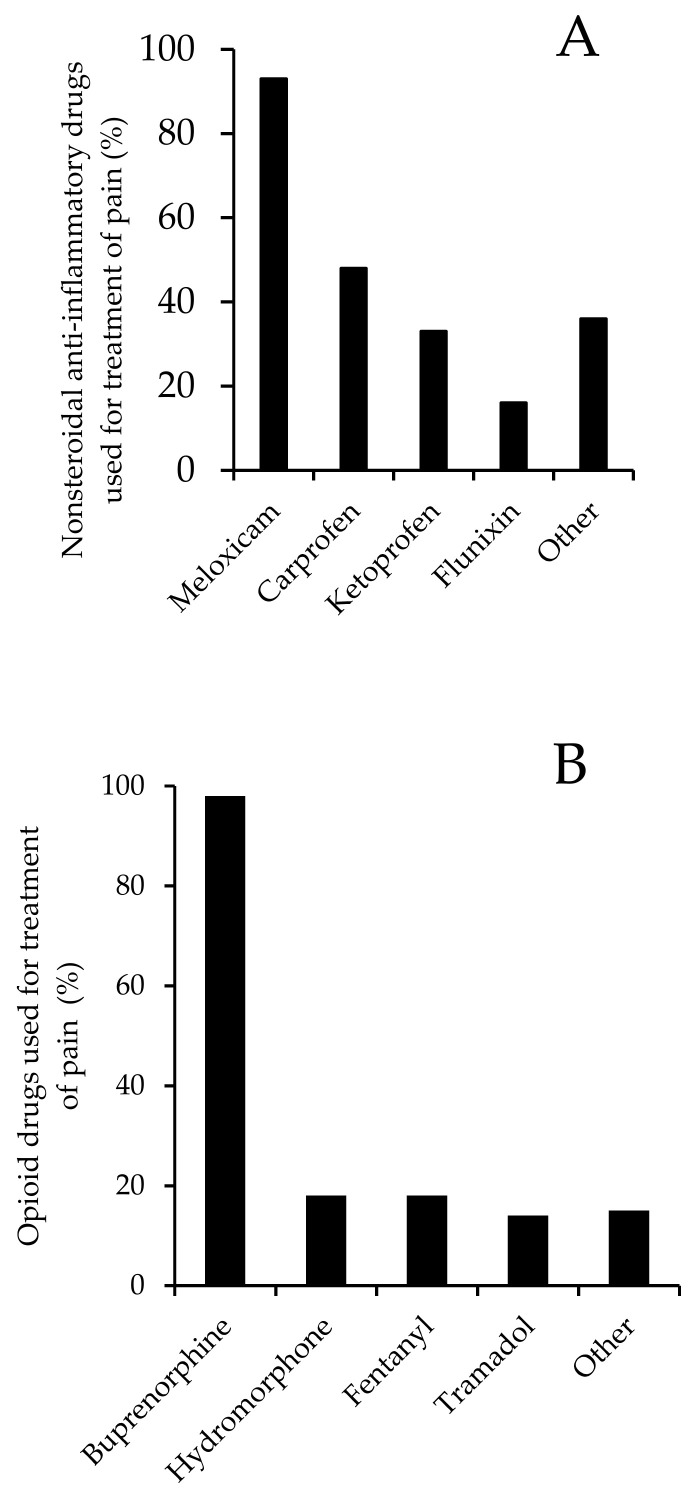
Pharmacologic agents used for the treatment of pain in primates (expressed as percentage of use): (**A**) NSAIDs (*n* = 83), (**B**) opioids (*n* = 82), and (**C**) local or topical anesthetic agents (*n* = 79).

**Figure 5 animals-12-02304-f005:**
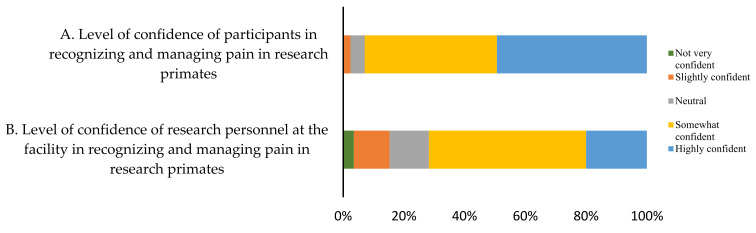
The level of confidence of recognizing and managing pain in research primates using a 5-score Likert scale (*n* = 85).

**Table 1 animals-12-02304-t001:** Demographics of primate pain survey respondents and primate species.

Parameter Assessed	No. (%) of Respondents	
Country		*n* = 92 ^a^
United States	83 (90)	
Canada	2 (2)	
Germany	2 (2)	
China	1 (1)	
Netherlands	1 (1)	
Barbados	1 (1)	
Puerto Rico	1 (1)	
Israel	1 (1)	
Institution type		*n* = 92 ^a^
University/Academic research	41 (44)	
Private or contract research	30 (33)	
Pharmaceutical research	7 (8)	
National Primate Research Center	6 (7)	
Hospital	2 (2)	
Non-profit/Sanctuary	2 (2)	
Primate breeding facility/Supplier	2 (2)	
Military	1 (1)	
Private consulting	1 (1)	
Primary job function ^b^		*n* = 91 ^a^
Veterinarian (Clinical/Attending/Research)	74 (81)	
Director	8 (9)	
Administrative	3 (3)	
Management	2 (2)	
Consultant	2 (2)	
IACUC member	1 (1)	
Post-Doctoral fellow	1 (1)	
Primary job includes working with living primates		*n* = 93 ^a^
Yes	89 (96)	
No	4 (4)	
Primate species currently worked with ^b^		*n* = 88 ^a^
Macaques	84 (96)	
Baboons	17 (19)	
Squirrel monkeys	16 (18)	
African green monkeys	12 (14)	
Chimpanzees	7 (8)	
Owl monkeys	6 (7)	
Sooty mangabeys	2 (2)	
Other	25 (29)	

^a^ Total number of survey participants = 93; ^b^ participants could select more than one answer.

**Table 2 animals-12-02304-t002:** Policies and protocols for research primate pain assessment and treatment.^a^

Parameters Assessed	No. (%) of Respondents	
Individual responsible for making pain assessment in primates ^a^		*n* = 85 ^b^
Veterinarian	83 (98)	
Veterinarian technicians	77 (91)	
Animal care personnel	54 (64)	
Other research staff	36 (42)	
Principal investigators	30 (35)	
Students	8 (9)	
Other	3 (4)	
Primates monitored after treatment to evaluate effectiveness of analgesia		*n* = 86 ^b^
Yes	78 (91)	
Sometimes	8 (9)	
No	0 (0)	
Frequency of unplanned top-ups in analgesic medication to primates		*n* = 85 ^b^
Often	3 (4)	
Sometimes	45 (53)	
Rarely	34 (40)	
Never	1 (1)	
N/A (no procedures requiring analgesic)	2 (2)	

^a^ Total number of survey participants = 93; ^b^ participants could select more than one answer.

**Table 3 animals-12-02304-t003:** Reported assays and methods used to recognize and evaluate pain in research primates.

Assay Category	Assay or Method	Description	Reference
Reflex-based	Application of noxious stimuli (i.e., chemical, thermal, or mechanic)	Dose-related increase in pain	[51,52,53]
Physiologic	Cage-side observation Thermometer/infrared thermography Stethoscope Urine, fecal, blood samples Telemetry	Blood pressure Respiratory rate or changes in respiratory rate Body temperature Heart rate Cortisol/adrenocorticotropin	[18,54,55,56,57,58]
Clinical	Cage-side observation Scale Quantify food intake	Body weight/body condition score Appetite	[59,60]
Behavioural	Cage-side observation Scoring grid Daily activity budget Ethograms Behavioural scoring using software (i.e., Observer XT)	General activity levels Posture Changes in species-typical behaviour Social behaviour	[19,20,61,62,63]
Facial expression	Cage-side observation (no validated grimace scale)	Pain grimace	[20]

**Table 4 animals-12-02304-t004:** Pharmacokinetics of common analgesics reported in primates.

Species	Class	Agent	Dosage	Route	Duration of Action	Cmax	Half-Life	AUC	Efficacy	Reference
Rhesus macaque	Opioid	Bupr	0.03 mg/kg	IM IV bolus	12 h 24 h	11.8 ng/mL C0: 33.0 ng/mL	- -	0–24: 1519 min*ng/mL 2188 min*ng/mL	No	[79]
Rhesus macaque	Opioid	Bupr (HCBS)	0.24 mg/kg 0.72 mg/kg	SC	48 h 72 h	19.1 ng/mL 65.2 ng/mL	α5.64 h β19.6 h α3.49 h β20.6	0–120: 236.4 ng*h/mL 641.3 ng*h/mL	No	[99]
Cynomolgus macaque	Opioid	Bupr	10 µg/h 20 µg/h	TDP TDP	5 d 6 d	3.3 ng/mL 8.1 ng/mL	64.2 h 42.4 h	0–168: 300.8 ng*h/mL 678.5 ng*h/mL	No	[100]
Cynomolgus and rhesus macaque	Opioid	Bupr	0.01 mg/kg 0.03 mg/kg	IM	6–8 h 12 h	8.1 ng/mL 40.7 ng/mL	2.6 h 5.3 h	0–120: 9.1 ng*h/mL 39.0 ng*h/mL	No	[45]
Cynomolgus and rhesus macaque	Opioid	Bupr-SR	0.2 mg/kg	SC	5 d	15.3 ng/mL	42.6 h	0–120: 177.0 ng*h/mL	No	[45]
Olive baboons	Opioid	Bupr	0.01 mg/kg	IM	12 h	-	-	-	Behaviour Heart rate Cortisol	[43]
Rhesus macaque	Opioid	Liposomal Hydr	2 mg/kg	SC	-	55.3 ng/mL	30.4 h	0–144: 424.7 ng*h/mL	No	[101]
Rhesus macaques	Opioid	Hydr	0.1 mg/kg	SC IV	-	26.4 ng/mL C0: 35.6 ng/mL	0.7 h 0.6 h	0–144: 32.5 ng*h/mL 36.3 ng*h/mL	No	[101]
Rhesus macaques	Opioid	Hydr	0.075 mg/kg	IM IV	2 h	12.0 ng/mL 77.6 ng/mL	81.5 min 17.7 min	- -	No	[72]
Rhesus macaques	Opioid	Fentanyl	1.3 mg/kg 2.6 mg/kg	TFS	7 d 10 d	1.95 μg/mL 4.2 μg/ml	90.9 h 97.4 h	0–504: 221.0 h/μg/mL 433.0 h/μg/mL	No	[102]
Rhesus macaques	Opioid	Fentanyl	0.005 mg/kg 0.01 mg/kg	SC	-	-	-	-	Thermode stimulation	[51]
Cynomolgus macaques	Opioid	Fentanyl	25 µg/h	TDP	4 d	2.4 ng/mL	45.2 h	0–96 h: 8.5 ng*h/mL	No	[98]
Cynomolgus macaques	Opioid	Fentanyl	1.95 mg/kg	TDS	4 d	177.1 ng/mL	32.8 h	0–96 h: 646.8 ng*h/mL -	No	[98]
Cynomolgus macaques	Opioid	Fentanyl	25µg/h	TDP	4 d	2.2 ng/mL	16.6 h	0–168: 110.3 ng*h/mL	No	[100]
Rhesus macaque	Opioid	Tram	1.5 mg/kg 3.0 mg/kg	IV PO	- -	C0: 540 ng/mL 15.2 ng/mL	111 min 133 min	- -	No	[72]
Rhesus macaque	Opioid	Tram	2.5 mg/kg 5 mg/kg	SC	-	-	-	-	Thermode stimulation	[51]
Cynomolgus macaque	NSAID	Mel	0.2 mg/kg 0.1 mg/kg	IM P0	24 h 8–12 h	2134.2 ng/mL 440.7 ng/mL	13.6 h 14.1 h	- -	No	[44]
Cynomolgus macaque	NSAID	Mel-SR	0.6 mg/kg	SC	48–72 h	3183.2 ng/mL	13.1 h	0–120: 80,407.4 ng*h/mL	No	[44]
Olive baboons	NSAID	Car	2.2 mg/kg	IM	12 h	-	-	-	Behaviour Heart rate Cortisol	[43]
Olive baboons	NSAID + Opioid	Car + Bupr	0.01 mg/kg + 2.2 mg/kg	IM	12 h	-	-	-	Behaviour Cortisol Heart rate	[43]

HCBS: highly concentrated buprenorphine solution; TDP: transdermal patches; TDS: transdermal solution; d: days; h: hour; Tram: tramadol; Mel: meloxicam; Car: carprofen; Bupr: buprenorphine; Hydr: hydromorphone: SR: sustained release.

**Table 5 animals-12-02304-t005:** Pharmacokinetics of analgesics used in other common research species.

Species	Class	Agent	Dosage	Route	Dosing Interval	Cmax	Half-Life	AUC	Efficacy	Reference
Dog	NSAID	Carprofen	25 mg	PO SC	12 h	16.5 μg/mL 8.08 μg/mL	4.95 h 7.07 h	0–12: 71.7 μg*h/mL 64.9 μg*h/mL	No	[108]
Goat	NSAID	Ketoprofen	3 mg/kg	IV	12 h	13.6 μg/mL	3.10	0–24: 7.71 μg*h/mL	No	[109]
Dog	NSAID	Ketoprofen	1.0 mg/kg	PO	-	2.02 μg/mL	1.65 h	0–12.5: 6.06 μg*h/mL	No	[110]
Rat	NSAID	(S)-Ketoprofen	3.2 mg/kg	PO	-	2.73 μg/mL	26.9 h	0–24: 34.5 μg*h/mL	Pain-induced function impairment	[111]
Sow	NSAID	Flunixin	3.3 mg/kg	TD	-	14.61 ng/mL	9.76 h	214.78 ng*h/mL	No	[112]
Mice	NSAIDS	Flunixin	2.0 mg/kg	SC	-	4553.4 ng/mL	0.95 h	0–6: 4742 ng*h/mL	No	[113]

## Data Availability

Dataset available from authors upon request.

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
