# Peer review of "Challenges with Assessing and Treating Pain in Research Primates: A Focused Survey and Literature Review"

_animals, 2022, doi:10.3390/ani12172304_

Round 1
Reviewer 1 Report
This is an important topic to address and the authors do an excellent job justifying the need for this information. The paper is generally well written.
The authors should make clear if the 93 respondents represented unique institutions in the abstract and summary. Throughout, it would be important to know if more than one respondent at a given institution provided information, as this would involve pseudo replication and could provide the opportunity for reliability checks on information about institutional practices.
Line 130, it is not clear what the two types of facilities are. Where do zoos fall? It does not seem that they are represented at all. Primates in lab settings are a special category since they are often subjected to painful or uncomfortable procedures and this practice should incentivize greater attention to symptoms of pain than might be typical in other settings. Perhaps the paper should be framed with this specific focus.
If respondents were asked only what species they currently worked with, what about species they had prior experience with?
Very few new world monkeys are represented.
The authors note that acupuncture is among the most common non-pharmacologic care used but this is not as shown in Figure 3.
I would be inclined to organize the paper in terms of first assessing pain, and then treating pain.
The discussion is unusually long. Perhaps the authors could cut some of the information about different drugs and their administration and focus more on strategies for assessing pain and the efficacy of medications. Most of the discussion is not directly linked to findings from the survey and seems more like a review of the authors’ opinions of techniques for pain assessment and treatment. This seems to veer away from the central topic. This information is useful of course but it might make sense to split this into two separate papers – one on the survey results and another on recommendations for treatment of pain in research primates. Perhaps I misunderstood the scope of the paper initially, believing it to be about the survey study, but now this seems like a small study that was folded into a review to provide a meatier paper. It is a bit unclear whether this is an empirical report or a review then.
I find it surprising that there are few institutions with pain protocols since this should be part of the IACUC review for any research related institution. Is it possible that staff is not aware of protocols followed by lead PIs?
I don’t know if Tables 2 and 3 are necessary.
There is a typo on line 12 p 1.
Line 262, guidelines is missing the s
Line 306 there is an extra “er” on “consider”
Line 300 is missing an “a” before “recent survey”
Author Response
The authors should make clear if the 93 respondents represented unique institutions in the abstract and summary. Throughout, it would be important to know if more than one respondent at a given institution provided information, as this would involve pseudo replication and could provide the opportunity for reliability checks on information about institutional practices.
Thank-you for your comment. We asked that only one response be submitted per institution but because this was a completely anonymous survey we can’t be certain how many institutions were represented in the end, only the total number of respondents.
Line 130, it is not clear what the two types of facilities are. Where do zoos fall? It does not seem that they are represented at all. Primates in lab settings are a special category since they are often subjected to painful or uncomfortable procedures and this practice should incentivize greater attention to symptoms of pain than might be typical in other settings. Perhaps the paper should be framed with this specific focus.
Thank-you for your query. The ‘two types of facilities’ refers to how the response was categorized if a respondent indicated that they worked for more than one facility.
While there are a small number of zoo vets who are APV members, this survey was specific to research primates. The paper is specifically geared to pain management in research primates and this is reflected in the title, abstract, introduction, etc.
If respondents were asked only what species they currently worked with, what about species they had prior experience with?
The purpose of the survey was to gather current data for assessments and treatments as protocols for treatment, techniques and other aspects of clinical care can change over time.
Very few new world monkeys are represented.
Yes, there are very few research facilities working with New World monkeys in research settings. This is similar to import data results reported annually by CDC.
The authors note that acupuncture is among the most common non-pharmacologic care used but this is not as shown in Figure 3.
Thank-you for your comment. Acupuncture is represented in the 6th line from the top. It was one of the most common non-pharm methods but not necessarily the most common method.
I would be inclined to organize the paper in terms of first assessing pain, and then treating pain.
Thank-you for this suggestion – we agree that this is more logical. We have switched Sections 3 and 4 to address this request.
The discussion is unusually long. Perhaps the authors could cut some of the information about different drugs and their administration and focus more on strategies for assessing pain and the efficacy of medications. Most of the discussion is not directly linked to findings from the survey and seems more like a review of the authors’ opinions of techniques for pain assessment and treatment. This seems to veer away from the central topic. This information is useful of course but it might make sense to split this into two separate papers – one on the survey results and another on recommendations for treatment of pain in research primates. Perhaps I misunderstood the scope of the paper initially, believing it to be about the survey study, but now this seems like a small study that was folded into a review to provide a meatier paper. It is a bit unclear whether this is an empirical report or a review then.
Thank-you for this comment. We thought it important to help frame the review by providing the results of this recent benchmarking survey. We hope that this spurs more interest and funding support for research into primate-specific pain assessment and management. We also thought it important to discuss the survey results thoroughly given that this is an area that the public thinks about a lot when considering the use of animals in research (i.e., are animals necessary, will it hurt them). Although there is a lot of empirical knowledge in the heads of primate clinicians, there are very few papers published on clinical management of pain in research primates and few peer-reviewed sources of information for clinicians and researchers. Like pain management in human patients, greater consistency in how objective assessments are conducted, combined with effective therapeutics with known duration of action will help to move the field forward.
I find it surprising that there are few institutions with pain protocols since this should be part of the IACUC review for any research related institution. Is it possible that staff is not aware of protocols followed by lead PIs?
We asked about generic vs primate-specific pain protocols. Most facilities had generic protocols but not primate-specific. This is important because pain features and approaches to assessments are not identical between species and these should be used as a basis for training. It’s also important to note that 100% of respondents indicated that they were monitoring primates after painful procedures/giving analgesic agents. So it is not a case that animals are not being examined or treated, more that there is a lack of consistency in what people are doing/using for assessments.
I don’t know if Tables 2 and 3 are necessary.
Respectfully, we think these are important. Table 2 because it is part of the survey summary and Table 4 (previous Table 3) because it summaries PK values for common analgesics in primates – really important for formulating the treatment plan. We are not aware of another published paper that has summarized this data to date.
There is a typo on line 12 p 1.
Corrected, thank-you.
Line 262, guidelines is missing the s
Corrected, thank-you.
Line 306 there is an extra “er” on “consider”
Corrected, thank-you.
Line 300 is missing an “a” before “recent survey”
Corrected, thank-you.
Reviewer 2 Report
The authors of this paper aimed at investigating the state of the art in assessing and treating pain in primates used for research. They do so by using both the results of a questionnaire, and the a review of the literature on the subject. The result is the need of increasing our knowledge on both aspects in order to arrive at a standardised methodology and guidelines to improve the quality of life of laboratory primates.
I am not a veterinarian, therefore I am not competent enough to evaluate the parts of the paper related to the use and efficacy of particular drugs to minimise pain in experimental subjects. Therefore, my comments will be more on general terms. I am confident that other reviewers will be able to cover those particular aspects as well.
Having said so, I found the paper very interesting, comprehensive and very useful for the laboratory community. I really applaud the efforts made by the authors. In my opinion, pending some minor revisions, the paper deserves publication in "Animals". Furthermore, the efforts by the authors is perfectly in tune with the concept of "Refinement", in the 3Rs philosophy.
Few comments:
2.4. Survey results: Can the authors specify how many questionnaires were sent?
3.: Is this where the literature review starts? Can the authors specify this?
3.5. Route of analgesic administration: 4th paragraph says "Intramuscular (IM) injection...direct handling (i.e., squeeze cages)." I dont understand this passage. The use of squeeze cages is becoming more and more obsolete, thanks to appropriate positive training techniques...can the authors comment on this?
I would move part 4 (Pain Assessment in Research Primates) BEFORE Part 3...I suppose assessment comes before treatment...
4.4. Behaviour: "Behavioural measures are considered the gold standard when assessing pain in laboratory animals". Is it so? Who said it? Is the authors' idea?
Can the authors elaborate on the possible use of vocalisations as a sign of pain and psychological discomfort in captive primates?
If the authors will address adequately these concerns this paper can surely by published in Animals. It is a significant contribution to a very important matter in the field animal experimentation.
Author Response
Few comments:
2.4. Survey results: Can the authors specify how many questionnaires were sent?
The surveys were sent electronically by the organizations via a member listserve, so presumably reached all members of the 2 associations. We used this information to estimate the response rate.
3.: Is this where the literature review starts? Can the authors specify this?
Thank-you for this suggestion. We have added ‘a review’ to the #3 and #4 headers to better identify the literature review sections.
3.5. Route of analgesic administration: 4th paragraph says "Intramuscular (IM) injection...direct handling (i.e., squeeze cages)." I dont understand this passage. The use of squeeze cages is becoming more and more obsolete, thanks to appropriate positive training techniques...can the authors comment on this?
Thank-you for your query. We have changed this to ‘requires minimal handling and restraint’ in the text. While we agree with the reviewer’s assessment that PRT should be more common it is still not common in most institutions in North America, particularly for veterinary treatments. Squeeze-back mechanisms are still in common use.
I would move part 4 (Pain Assessment in Research Primates) BEFORE Part 3...I suppose assessment comes before treatment...
Thank-you for this suggestion – we agree that this is more logical. We have switched Sections 3 and 4 to address this request.
4.4. Behaviour: "Behavioural measures are considered the gold standard when assessing pain in laboratory animals". Is it so? Who said it? Is the authors' idea?
Thank-you for this comment. We have tempered the stated enthusiasm for behavioural assessments and deleted the entire sentence.
Can the authors elaborate on the possible use of vocalisations as a sign of pain and psychological discomfort in captive primates?
A comment was added that ‘vocalization in response to touch or palpation’ may be indicative of pain.